# Dirty South Feminism: The Girlies Got Somethin' to Say Too! Southern Hip-Hop Women, Fighting Respectability, Talking Mess, and Twerking Up the Dirty South

**Adeerya Johnson**

The Social Justice Institute, The University of British Columbia, Vancouver, BC V6T 1Z1, Canada; adeeryaj@gmail.com

**Abstract:** Within southern hip-hop, minimal credit has been given to the Black women who have curated sonic and performance narratives within the southern region. Many southern hip-hop scholars and journalists have centralized the accomplishments and masculinities of southern male rap performances. Here, dirty south feminism works to explore how agency, location, and Black women's rap (lyrics and rhyme) and dance (twerking) performances in southern hip-hop are established under a contemporary hip-hop womanist framework. I critique the history of southern hip-hop culture by decentralizing male-dominated and hyper-masculine southern hip-hop identities. Second, I extend hip-hop feminist/womanist scholarship that includes tangible reflections of Black womanhood that emerge out of the South to see how these narratives reshape and re-inform representations of Black women and girls within southern hip-hop culture. I use dirty south feminism to include geographical understandings of southern Black women who have grown up in the South and been sexually shamed, objectified and pushed to the margins in southern hip-hop history. I seek to explore the following questions: How does the performance of Black women's presence in hip-hop dance localize the South to help expand narratives within dirty south hip-hop? How can the "dirty south" as a geographical place within hip-hop be a guide to disrupt a conservative hip-hop South through a hip-hop womanist lens?

**Keywords:** hip-hop womanism; dance; southern studies; black feminism; hip-hop studies; southern hip-hop; dirty south feminism; black girlhood studies; respectability politics



## 1. Introduction

When it comes to the dirty south, the place-based hip-hop subculture includes dance/party-based music, skating rinks, strip clubs, luxury Cadillac cars, and often objectifying women through pimp personas and misogynoir with hypersexual lyrics. However, through the perspective of women rappers, the narrative centers around self-love of the booty, asserting sexual desires, representing, claiming their perspective cities, confidently speaking about the value they bring into the rap game, and lyrical geniuses. Confidently rapping over a bass-heavy southern beat with a femme flair and bad bitch persona expands the reputation in which Black women connect to the dirty south.

The phrase "dirty south" is defined as a subculture of hip-hop music emerging in the 1990s. The sub-genre highlights southern-based cultures such as Atlanta/Memphis Crunk, Miami Bass, New Orleans Bounce and Houston' Chopped and Screwed that embody southern hip-hop sonic productions. The dirty south was unique in providing a particular sound and cultural experience that was distinctively different from other hip-hop regions. The earliest usage of the term came from the Atlanta-based rap group Goodie Mob, from their song titled "Dirty South". The song presented a new culture of observing hip-hop in the American South through the perspective of Black men that included a new understanding of southern culture, history, and socio-economic contexts (Miller 2008). The

geographical uniqueness of dirty south hip-hop was mainly in the accents, fashion, lyrical content and musical production.

Music from the dirty south included the ways in which rappers, DJs and dancers composed sounds, songs and imagery that included a southern drawl, laidback ambiance, heavy-bass, simple call-and-response lyrics, booty/hip movement and chants that often included sexually oriented themes. The foundation of the musical and cultural traditions that make southern hip-hop different regionally is the southern economics with ties to slavery, carceral history and modernity that shapes Black southern identity, which emerges not only in the music but also all five of the pillars of hip-hop. This appears in the southern landscape and speaks to the urbanity and rural spaces that occupy the culture and conversations that emerge in the commentary and expression of southern hip-hop artists (Barnes 2020; Bradley 2021; Grem 2006; Miller 2004, 2008; Nichols 2014; Robinson 2014). Importantly, the physical locations in which parties, events and social gatherings take place in the South have important connections to knowledge building and the many ways in which Black women conceptualize their southern experience. I argue that these experiences of Black women, and even Black girls, serve as a marker and a way of mapping southern hip-hop culture. Therefore, Black women fuel the economic, social and cultural politics of the South through dance and rapping which challenge gender, class and representational politics within southern hip-hop culture. In this paper, I argue through the literature surrounding dirty south feminism that Black women and girls disrupt respectability politics within southern hip-hop while also providing sexual, economic, and political support through their lyrical and performative contributions.

## 2. Dirty South Feminism and the Women of the South

With the rise of southern hip-hop music, Black women were able to tell their stories in the mid-90s, with artists emerging out of cities like Portsmouth, VA in 1991 with Missy Elliott; New Orleans, LA in 1994 with Mia X; and Memphis, TN in 1998 with Gangsta Boo and La Chat. However, the maturity of southern hip-hop culture has failed to acknowledge the accomplishments and uniqueness of Black female rap artists, DJs, and dancers as significant contributors as southern lyricists, producers and performers. In contemporary hip-hop and traditional Black feminist studies, Black women's engagements in southern hip-hop have been critically pathologized, victimized and loosely explored. Scholars have classified Black women in hip-hop as modern day jezebels and sapphires who lack agency and control due to misogynistic images and heteropatriarchal structures in the industry (Brown et al. 2013; Cole and Guy-Sheftall 2009; French 2013; Lindsey 2015). Ultimately, this singular narrative achieves two flawed goals; erasing the possibility of agency and experiences of Black women and girls, and introducing notions of respectability that formulate in the South. However, hip-hop feminist scholars highlight the social and political contradictions in hip-hop by underlining Black women's abilities to have agency over their bodies, lyrical themes and sexual desires while challenging misogynoir in hip-hop.

Hip-hop feminist writer and journalist Joan Morgan (2015) poses the question, "What possibilities can a politics of pleasure offer for Black feminist futures?". She asks this in order to explore the range in which Black women can affirm and centralize notions of pleasure under a Black feminist framework (Morgan 2015). Roach (2019) explores similar notions of Black women's sexual freedoms and explores alternate possibilities of Black women's sexual pleasures throughout the positionality of invisibility, interiority, domesticity, and privacy. In the presentation of hip-hop feminist ideology, Morgan (1999) expands this conversation and discourse in the ways in which Black women can eliminate conceptions of being seen as victims in hip-hop culture and assert their experiences and allegiance to hip-hop as a significant part of their lives. The extension of this literature has been supported in conversations within Black girlhood studies about Black girls' creative potential, and the ways in which Black girl dance is diasporic, which I argue is significant in the ways in which they internalize their southern dancing experiences. I ask how these

notions can be explored under the discipline of dirty south feminism and be applied to new representations of Black women and girls in southern hip-hop that support agency and creativity? Furthermore, how do southern Black women and girls challenge respectability politics and heteronormative patriarchy within Southern hip-hop culture and present nuanced hip-hop feminist narratives that emerge in southern hip-hop dance politics?

Dirty south feminism works to centralize the importance of Black womanhood and girlhood through performances of twerking and booty shaking, and southern lyrical themes that situate understandings of the agency and lived realities of Black women who engage in southern culture. This exploration provides insight into autonomy and personhood. Dirty south feminism acknowledges southern Black girlhood and locates Black girl creativity. Dirty south feminism highlights the narratives of Black queer women and Black femmes to investigate how the impact of gender and sexuality within the South has influenced understandings of Black femininity, masculinity and androgyny through music and dance. The scholarship lends itself to southern Baptist households, and in that it addresses both Black church politics and the disruption of respectability politics. The underlying theme of dirty south feminism works to explore and uplift what southern Black womanhood and girlhood means while studying the impact of race, class, religion, gender and sexuality that emerges through performances of southern hip-hop dance cultures. The collection of dirty south culture, performances and sounds is a part of a socio-cultural experience that formulates relevant connections to a specific southern hip-hop sound and a social function in which Black women and girls find value and representation. The work of dirty south feminism provides a contemporary and southern perspective of Black women and girls that was missing in the exploration of early hip-hop feminist works and southern hip-hop studies analysis.

### 3. Southern Girl and Atlanta

With the growth of southern hip-hop music, Black women have been able to establish their voices in the subgenre with artists like Miami-based rapper Trina and New Orleans's rapper Mia X. While these women gave rise to Black women's understandings of the South, they were able to formulate contemporary prodigies such as Houston's Megan thee Stallion, Miami's City Girls and New Orleans bounce artist Big Freedia. These artist support hip-hop feminist scholars' argument for a feminism that "fucks with the greys" while addressing nuanced representations of Black women as a part of the hip-hop generation (Cooper et al. 2017; Lindsey 2015; Morgan 1999). In the dirty south feminist context, southern rap artist Megan Thee Stallion often represents being a Houston, Texas native and adorns herself in sexy cowboy-like attire. Megan has established a southern hip-hop style and sound in which she mixes her confident and assertive rap persona with southern influences from rappers like Pimp C and Three 6 Mafia and hometown Texas culture through her fashion. Megan also establishes an authentic sound in which Black women who are fans can connect to her music due to its applicability in reaffirming their own sexual identities, southern upbringing and desires related to, first, their identity as Black women and, second, their position in the South.

Unfortunately, this extension of representation and idea of place lacks in the recognition of the lived realities of Black women from Atlanta in hip-hop. I find this happens due to the centralization and importance of male experiences in hip-hop that is dominated by men. The lack of representation is an act of misogynoir and disregard to Black women's marketability in hip-hop, a lack of seriousness and quality southern hip-hop music compared to East and West Coast rap, and the ignorance of Black women and girl's experiences, impact and relevance to southern hip-hop culture. Additionally, the exploration of southern hip-hop in Atlanta is necessary to extend the literature of the South as an artifact of Black girlhood performance through twerking and the ratchet imagination that has been established by Black women and Black girls across the South. This includes the particular material and behavioral establishments in southern Black cultures, as represented from the lived experiences of Black women growing up in cities like Atlanta. The earliest expressions

of Black girlhood performances in support of the Atlanta hip-hop scene are in the extension and inclusion of a variety of performances in Atlanta's Spring Break party known as Freaknik. Curated by Atlanta's HBCUs, Black women produced, transformed and defined cultural interactions within this street party (Thompson 2007). The participants of Freaknik in Atlanta were Black collegiate youth who were key components of booty shaking and twerking performances that occurred in the streets of Atlanta by Black women (Halliday 2020; Thompson 2007). The annual event specifically highlighted how Black women could shape southern culture through dance and street performance. This is evident in the ways in which Freaknik was the backdrop to a few Atlanta-based southern music videos such as Uncle Luke's 'Work it Out' video at Freaknik 1993 and Playa Poncho and La Sno's 'Whatz Up, Whatz Up' video at Freaknik 1995. Even Atlanta producer Jermaine Dupri curated a compilation album called *So So Def Bass All-Stars* with various southern hip-hop artists taking on the Miami and Atlanta bass sound to produce Freaknik-style music.

Through the rise of the crunk era in Atlanta after the end of Freaknik, Diamond and Princess from the rap group Crime Mob and solo-rapper Rasheeda were the first to emerge; they introduced a new era of southern Black womanhood and performance in Atlanta. Diamond and Princess have established a place in the Atlanta hip-hop scene and nationwide with their notable verses in 'Knuck If You Buck'. The song peaked at number 75 on the US Billboard Hot 100 and number 23 on Billboard Hot Rap Songs, which ultimately redefined Atlanta party culture (Nelson 2019). At the time the duo emerged, they reflected the identities and style of Black women living in Atlanta with their songs like 'Stilettos' and 'Rock ya Hips' tapping into Black women's fashion and body politics (Nelson 2019). Most importantly, Diamond and Princess made their claim to fame in southern hip-hop through the intersections of race, class and sexuality. This means considering how Diamond and Princess were one of the first to introduce people living outside the South to Black girls living, rapping, and dancing in Atlanta. With their urban southern style, rhymes, and aesthetic, Diamond and Princess represented the everyday look and experiences of Black girls from the crunk era in southern hip-hop. Thus, by highlighting experiences of southern women rappers in hip-hop and Black feminist studies we can explore the ways in which disrupting conservative and patriarchal southern cultures shows how Black women as artists, performers, and consumers of hip-hop have always been foundational to the construction of southern hip-hop music and culture.

## 4. Battling the Heteronormativity and Respectability

The dirty south and its geographical relation to southern cultures within hip-hop determines how members of the South navigate between a Black conservative South and a heteronormative music genre (Love 2017; McKittrick 2006; Miller 2008). Southern hip-hop culture is most connected to representations of Black manhood and hypermasculinity. This is seen in the strength of pimp culture, sexist rhetoric and homophobic street culture often reflecting micro-systems of power and dominance within Black communities that are not granted to Black men through white patriarchy (Love 2017; Miller 2004). Southern hip-hop is supported here in the ways in which performances within hip-hop incorporate actions and feelings about masculinity, power and asserting manhood (Balaji 2012). The position of Black women's sexuality in hip-hop has been situated in the ways in which men in hip-hop speak about how women receive sex and sexual objectification. However, how do Black women navigate misogynoir in the South, but also recognize and engage with sites such as pimp culture, power and dominance?

With the emergence of southern rap groups, artists made it important to insert at least one female rapper into their entourage. The spot to be the only woman in the group was significant, and was marketed as a standout female performer—which is seen in southern rap groups like Three 6 Mafia, Slip-N-Slide and No Limit Records. The inclusion of Black women rappers is a significant act of visibility to provide a small extension to the South by disrupting normative understandings of southern Black womanhood and southern rap groups' gender dynamics. The inclusion of female rappers in the rise of southern

hip-hop reshaped notions of the "southern belle" due to their own intersections of race, class and gender in these southern cities. Specifically, New Orleans-based rapper Mia X and member of No Limit Records refers to herself as an "unladylike diva", but also a "lyrical man eater," in her feature on Master P's 'Make Em Say Uhh!'. We see here this disruption of the southern belle and respectability politics in the southern context as Mia X challenges heteronormativity with her aggressive lyrics asserting that she has the lyrical strength to compete against other men in the rap industry. This signifies the reality that Black womanhood has a specific identity in southern hip-hop that supports the region's socio-cultural roots and juxtaposes misogynoir.

Often, misogynoir has impacted sexual narratives, making topics about sex homogenous, and has not made room for women to explore their sexual pleasures and desires as the dominant partner. Similarly, the process of exploring Black women's sexuality outside of heteronormative frameworks makes room for queer narratives in hip-hop culture and new language for Black women's sexual lives. With many of the historical ties that examine Black women as victims of sexual assault, stereotypes and sexual scripts, forms of discomfort found in Black women's sexual narratives in hip-hop assert the possibility that Black women and Black queer/lesbian women like to "fuck" (Bailey 2019; Morgan 2015; Roach 2019). This assertion of sex positivity and body positivity challenges many valued beliefs and assumptions in the Black community about race, sex and gender politics. When Black women can shape and perform their own notions of pleasure in hip-hop, it ultimately disrupts patriarchal structures in the hip-hop industry, establishing new language and possibilities for Black women to address their sexual desires and defy respectability politics. Furthermore, this concept supports the chance for individual and collective social transformations in southern hip-hop culture regarding Black women's sexual desires coming out of the conservative anti-Black south.

There has been significant research about the ways in which slavery has impacted the notions of physical Blackness and Black sexuality under the white male gaze (Bailey 2019; Maxwell et al. 2015). This includes moving forward from conversations about hypersexualized images and stereotypical sexual scripts such as the jezebel or sapphire in Black feminist discourse that support respectability politics. Sticking to these notions of Black women's sexuality, southern hip-hop research lacks in providing the potential for Black women to explore their own understandings and possibilities of pleasure, eroticism and desire (Hunter and Cuenca 2017; Morgan 2015). Therefore, respectability politics impact the very notion of sex, pleasure and desires as forbidden opportunities that leave Black women with the only option to perform sex in a heteronormative and southern Baptist context under the pretense of marriage, virginity or private sex lives under the submission of men (Cole and Guy-Sheftall 2009; Durham et al. 2013; Morgan 1999, 2015). The legacy of women in hip-hop lives in a binary of being a respectable woman versus being a hypersexualized jezebel. Scholars have also noted in contemporary hip-hop that Black women are now battling the binary of respectability and the ratchet stereotype (Harris 2003; Love 2017). Recurring images such as the Gold Digger, Video Girl, Freak, Dyke and Baby Mama are some of the many sexual scripts that are prevalent in shaming Black women in hip-hop culture (Ross and Coleman 2011; Stephens and Few 2007). However, the research fails to understand the possibility that those scripts represent the real lives of real women that live within the intersections of race, class, gender and sexuality.

Disrupting respectability in southern hip-hop brings a new challenge, considering that southern hip-hop has a large strip club culture that survives in the conservative south. While Black women's sexual agency and revealing fashion disrupts images of the poised and respectful southern woman, female rappers are going up against patriarchal southern American values and social constructs of Black manhood in rap music. Many hip-hop feminist scholars have pushed for the end of respectability politics and the deconstruction of notions of gender critique and shame (Harris 2003; Pickens 2015). This is evident in the way that hip-hop has a long relationship to the production of pornography in the early 90s coming at the same time that we saw the birth of southern hip-hop. However, southern

conservative notions of sexual representations within the southern landscape amongst the hip-hop scene must include its connections to sex work within the cultural significance of strip club culture. This process also disrupts and redefines the southern pimp–hoe culture that disadvantages sex workers in the South.

Contemporary hip-hop has profited from visuals of Black women's bodies due to their sexually seductive and erotic performances (Cole and Guy-Sheftall 2009; Miller-Young 2014). Scholars have discussed Black women's sexual agency, where they have the power to sell the rapper with their dancing abilities and looks (Bailey 2019; Miller-Young 2014). This signifies a critical change in the representations of the Black women's sexuality and gender relations in pornography and sex work. Thus, this process redefines Black women's sexual labor, which is often shamed in traditional Black feminist discourse. Therefore, it is important to include the narratives of Black female strippers in this study to highlight the many ways in which Black women enter these industries because they find erotic performances offering opportunities that are not available in other kinds of sex work (Bailey 2019; Miller-Young 2014). With the insertion of sex, Black women in southern hip-hop cultures are able to define pleasure politics and renegotiate their sexuality on their own terms. Here, Black women can use their sexuality as a tool of resistance against heteronormativity and respectability while simultaneously rejecting misogynistic images that have been used historically to control and victimize them.

## 5. Sexual vs. Non-Sexual Dancing

The conflicts of Black women's sexuality within southern hip-hop are grounded in the fight against racial and historical stereotypes that developed in the antebellum south during slavery and manifested through Black minstrels, which ultimately impacted societal constructs. Organizations like the Black Women's Social Clubs and Black Greek Letter organizations in the early 20th century worked collectively to combat social stereotypes of being hyper-sexual, masculine, or labeled as a mammy, jezebel or sapphire (Collins 2005; Davis 1999; Pough 2015). Furthermore, with the growth of sexual politics that materialized in the 90s and early 2000s hip-hop culture, the emphasis on sex, money and power further complicated the perceptions and mainstream stereotypes that permeated the participation of women in hip-hop. With imagery of Black women as video vixens, strippers and dancers, the identities of the gold digger, baby momma, freak, groupie and video ho positioned a new narrative of Black women in hip-hop. This translated into the notion that Black women in hip-hop were victims of the genre, lacked agency, and were solely seen as props to the male rappers to be objectified (Durham et al. 2013; Hollander 2013; Love 2012, 2016; Neal and Forman 2004; Peoples 2008; Sharpley-Whiting 2008).

While these perceptions of Black women in hip-hop have truth to them regarding the sexually violent and hyper-masculine lyrics that impact the gendered power dynamics in hip-hop, many hip-hop feminists have explored the possibilities and realities that Black women have agency within hip-hop regarding their sexual performance. I argue there must be more conversations on the ways in which Black feminist scholars discuss how Black women explore their sexuality and pleasure that do not assume that they are victims to sexual expression or that it is absent of southern hip-hop experiences. Morgan (2015) interrogates ethnic heterogeneity, queerness, digital technologies and social media, which I argue are all important components of contemporary southern hip-hop dance culture. While Morgan offers a perspective within her scholarship that addresses the intersections that Black women have to unravel while exploring pleasure, I assert that southern Black women and girls' performances of hip-hop dance offer new ways for Black women to express themselves physically through dancing. They participate in a genre that not only advocates for a hustler, go-getter mentality but also as a dance culture that features Black women and girl's expressive cultures through identity, pleasure and body politics.

Within southern hip-hop dance, one of the major tensions of Black women's public forms of sexual performance appears within the strip club and music videos as dancers. Black women who occupy hip-hop's visual erotic dance cultures actually center and

support the lives of the Black sex workers that fuel the sexual economy of hip-hop (Miller-Young 2014; Sharpley-Whiting 2008). To specify, this explores the acts of selling sexual fantasies, strip teases, exotic dancing, pole work and pussy popping as forms of southern hip-hop entertainment. Their agency and performance of these dances includes the ways in which sex work within southern hip-hop as strippers and video vixens presents new power dynamics in which Black women shape erotic representation and labor politics within southern hip-hop dance culture. Scholars validate this notion of creative power and erotic agency where Black women in hip-hop use sex and sexuality as currency, which is more blatantly stated by Morgan as exploring "honest bodies that like to fuck" (Bailey 2019; Clay 2007; Miller-Young 2014; Morgan 2015).

Collins (2005), on the other hand, is hesitant to explore the differences in representation of sexual liberation and women who are victims of sexual objectification within hip-hop culture. I—along with Sharpley-Whiting (2008), author of *Pimps Up, Hoes Down*—suggest that we talk and listen to the strippers, video vixens and groupies who occupy those sexual dance economies within hip-hop culture. I find that extracting interviews from strippers and video vixens like Karrine Steffans (famously known as "Superhead"), Melyssa Ford or Buffy the Body grants them a space to discuss their career choices, and to address Black women's sexual and physical freedoms that challenge male notions of female sexuality and pleasure. Ultimately, the inclusion of sexual and more physically erotic dances tied to sexual fantasies includes the perspectives of Black women who occupy new Black sexual politics that appear in hip-hop dance culture. The interrogation and praxis of including Black sex workers in southern hip-hop culture caters to the Black feminist and hip-hop feminist futures of incorporating nuanced sexual politics surrounding the agency of Black women's bodies, which have ties to African diasporic erotic dance expressions found in Caribbean and African cultures (Halliday 2020; Morgan 2015; Sharpley-Whiting 2008). Therefore, it is important to emphasize that Black expressive dance cultures are Afrodiasporic, if one explores the presence and movement of the booty/butt in African and Caribbean dance cultures that are unique to the South (Defrantz 2016, 2018; Gottschild 2005; Halliday 2020).

The extended conversation on Black women's inclusivity of southern hip-hop dance and dirty south feminism explores Black girls' creative potential, which often appears as the performance of twerking, Black girl play and majorette dancing. Similarly to the former conversation, there is an attachment to respectability and modesty in the nature in which Black girls choose to express themselves through southern hip-hop dance. The cultural and social concerns within Black communities that aim to protect the lives and wellbeing of Black girls aim to protect them from controlling images and historical stereotypes that are found within hip-hop culture. However, there is a flaw that fails to acknowledge the different cultural and historical impact of social dancing in the South as a key component of southern Black expression. Two of the pillars of hip-hop are dance and knowledge of the self, which I argue are related themes that help Black girls navigate pleasure and self-expression. Similarly, many Black girlhood and hip-hop feminist scholars agree that Black girls' kinetic orality caters to their ability to build community and self-expression, and unveil creative and intellectual potential through Black dances like twerking and majorette dancing (Brown 2013; Gaunt 2006; Halliday 2020; Lindsey 2013; Love 2012, 2017). Black women and girls build knowledge through embodied experiences in which their connection to hip-hop dance provides a nuanced and inclusive look into the ways in which their socio-cultural and political ties to hip-hop culture serve as a sight for knowledge making, creativity, and self-expression. Examples of performances of southern Black girls' expression and creativity include Tik Tok dancer Jalaliah Harmon, the cast of Black majorette dancers in the Lifetime tv show *Bring It!,* and many other young girls who have publicly danced for HBCU bands, double-dutched in their local neighborhoods, and perform the latest dances in southern hip-hop music videos. Therefore, Black girls' pleasure within southern hip-hop dance is not solely tied to sexual intimacy but also to the nature of joy, laughter and pleasure that emerges from Black girl play and creativity in the South through twerking and hip movement. The sexualized nature of twerking or any

form of pelvic thrusting that emerges within hip-hop dance culture comes from how white hegemony and white mainstream culture has socially categorized all Black physical bodies as hypersexual and freakish (Bailey 2019; Collins 2005; Gottschild 2005). Centralizing on the claim of hypersexuality limits the nuance and the impact that Black girl creativity and expression have on the cultural formation of southern hip-hop.

The contemporary and more American translation of booty shaking in hip-hop culture that is practiced by Black women and girls presents a new meaning of performativity, and becomes inclusive of the ways in which they define many of the social contributions of hip-hop dance that emerge through local dance cyphers and digital spaces via social media. The southern performances of Black dance that centralize on the butt that appear in popular social dances or majorette HBCU dancing are rooted in Black dance traditions that reclaim the Black female/non-binary body (Defrantz 2016; Gottschild 2005). Through Black women and girls' public performance and lyrical themes of twerking and other non-sexual performances within southern hip-hop they are able to promote their culture. Furthermore, the usefulness of Black girl's dance cyphers pinpoints new relationships between Black women and Black girls to renegotiate private and public relationships with their bodies. Through performances of rapping, social dancing, twerking and Black girl play, the scope of inclusivity redefines the approach to explore hip-hop feminism and dirty south feminism through a performance lens. The cultural and social development of Black girl dance performances presents material realities in which movement and memory are constructed from within Black communities that appear in southern hip-hop dance cultures. This situates the reality that Black girls are aware of their connection to southern hip-hop culture and shape their narratives and existence similarly to Black women rappers, but also defy the hegemonic and European standards of dance and beauty that emerge in Black dance traditions.

## 6. Conclusions

Black women disrupt respectability within southern hip-hop while supporting the sexual, economic, and geographical politics of the dirty south through the public sphere, bringing wreck and southern performances and culture. My understanding of this phenomenon through dirty south feminism is grounded in the process of thinking about the socio-political and cultural space of southern hip-hop expression and identity in which Black women and girls in the South use southern hip-hop as way to explore the gendered politics of the South, and southern racialized identity, body and sexual politics. Black women and girls have disrupted respectability politics in the South sexually through the public sphere by taking up geographical space in the South through public social dancing. The sexual politics that emerge support call and response music, booty shaking and the sexual intercourse themes that are present within southern hip-hop music. The physical locations in which parties, events and social gatherings take place in the South have important connections to knowledge building and the ways in which Black women conceptualize their southern experience that I argue locate and map southern hip-hop culture. Black women also fuel the economic politics of the South through monetization and the physicality and musical embodiment of the trap, which challenge gender, class and representational politics within southern hip-hop culture, which ultimately still has ties to the sexual politics of the South. Additionally, the economic confounds that are present within southern hip-hop culture are the impact of Black women sex workers and how public sexual entertainment supports the economic growth of not only male rappers but also women rappers in the South who seek to use strippers or accentuate strip club culture in their music. Additionally, the hustler culture that emerges out of trap music, which is a subgenre of southern hip-hop music, also presents the discussion of Black southern women discussing and contributing to an illicit lifestyle that develops from the displacement of Black urban and rural life. Furthermore, the geographic mapping of Black women's life and their understanding of southern hip-hop culture has connections to the way in which Black women build a cultural understanding of beauty standards, southern values, gender roles

and dialect that is attached to multiple southern locations (McKittrick 2006). Therefore, these markers of identity and representation that define the attributes of Black women in the South appear in the style and fashion of southern hip-hop, Black girl aesthetic and creativity. I want to highlight this also as the different personas that occupy Black women's lives in the South. These identities are often labeled as the southern belle, the country girl, the city girl, and the trap girl. With these identities I ponder on Black southern identities and how the presence of the women who make up the majority of the working class in the South create their own hip-hop culture and economy based on their location in both urban and rural southern cities.

The Crunk Feminist Collective and their scholarly efforts center the lives of Black women and girls through a hip-hop feminist framework to disrupt and occupy the socio-cultural and political connections to hip-hop through the practice of "crunk". However, their emphasis on collaborating on the southern genre of crunk music originating from Memphis and Atlanta with their hip-hop and Black feminist analysis falls short of centering Black southern life from the perspective of Black women and girls. Their ability to address disruption, agency and respectability politics does not consider the sexual and economic structures that emerge from crunk music, trap culture and southern urban and rural life. However, Pough (2015) argues that attention must be drawn to the public sphere, in which music can be a space for women to dominate and control the public discourse in order to better explore the Black expressive cultures of Black women. She mentions that this has happened in the past because of Black blues women who discuss sexuality, female dominance, lesbianism, female erotic desires, and disrupting classed and racialized notions of love and sexuality (Davis 1999; Pough 2015; Rose 1994). We see this with the emergence in southern hip-hop, in which Black women in the South craft the public sphere by what Pough calls "bringing wreck". Therefore, because southern hip-hop has historical and cultural ties to dancing and call and response music, the dances that are mapped out across the South that are read as hypersexual or misogynist have a level of cultural capital that is passed as a shared southern social tradition.

The experiences that materialize from the South, kinetic orality through Black girl play, social dancing, and expressive cultures such as musical Blackness between the sexes has unique connections to the growth of southern hip-hop culture and politics of the South. Bradley (2021) further explores this concept, and argues that Black girls from the South navigate southern respectability politics, but through what she calls country Black girl essentialism. She describes this as the ways in which southern Black girls assert themselves into a culture and space to establish their own voice and identity that happen outside of southern maleness, the Black church, and the politics of southern respectability of being ladylike or proper (Bradley 2021; Davis 1999; Graham 2017; Robinson 2014). To return to Pough, this means looking at how Black women and girls in the South bring wreck through speech patterns such as southern dialect, slang or call and response chants, which can be performed as dancing or showing out in public. The particular performances in the South that support this notion are seen in hand games Black girls play, Black Greek organizations and majorette dancing found amongst historically Black colleges and universities, which are mostly found in the South. Furthermore, artists like Megan thee Stallion bring wreck with their southern presence with twerking, cowboy hats or cow print outfits, and she dominates the South with her Texas roots, where she discusses her influence from Houston rapper Pimp C and her late mother (Fitzgerald 2019). Megan also uses rap alter egos—The H-Town Hottie, Thee Stallion, and Tina Snow—and embodies twerking and stripper culture in her music and videos to signify her Texas roots. Importantly, the process of memory and storytelling in the South by Black women adds to the mapping of southern experiences. This also appears as community intelligence, which caters to the notion of a homeplace and the act of southern home training, with trapping or trap storytelling, as Black women build the narrative as a hustler in their community (Bradley 2021; Fitzgerald 2019; Miles 2020). Southern rappers like Jucee Froot, Latto, City Girls and Light Skinned Kiesha embody the trap girl persona, negotiate gender performativity, and localize their surroundings of

hustling, scamming and stealing to survive their marginalization. Additionally, exploring trap through a dirty south feminist perspective in order to explore respectability and disruption in the South highlights the narratives that emerge from trap as a creative epistemological genre that resist objectification, promote agency and engage in nuanced stylistic choices that are told by Black women from poor and working class communities (Miles 2020). To be specific, this critical analysis of trap supports Miles (2020) theory of trap feminism, which explores how Black women trap artists from the South create spaces that disrupt and respond to gendered racial capitalism through heavy bass music. Black women who occupy the physical and musical terrain of the trap are foundational to marginalization and modes of survival, and the methods Black women take to protect themselves and family. The unfortunate downside is that the performances, lyrics and actions that are products of the trap or trap music translate as unladylike, degrading and selfish. Furthermore, engaging in the politics of trap music and trap feminism explores how Black women that participate in the subgenre validate southern hip-hop musical traditions, construct meaning and commentate on their socio-political surroundings as marginalized subjects (Bradley 2021; Jennings 2020; Love 2017; Miles 2020; Rose 1994).

The public sexual terrain of southern hip-hop through the era of Freaknik, New Orleans Bounce music and southern strip clubs serves as a public display of sexuality and cultural expression. These particular southern performances and events disrupt respectability politics within the South that centralize as othered southern culture and performance which can be read as dirty (which is key to the dirty south) (Grem 2006; Horton-Stallings 2020). I want to note that these particular performances are just a small example of the many performances of southern hip-hop life that are practiced by Black southern women. The sexual performances and emphasis on the southern booty have very specific economic and geographical ties to the musical production of southern hip-hop music. I find that southern hip-hop texts fail to emphasize or imply that the music is not only to bring Black communities together but to find the perfect sonic sound to get Black women and girls to dance. The presence and growth of southern hip-hop acts in cities like Miami with 2Live Crew, 95 South, 69 Boyz and Slip and Slide Records, help creating Miami Bass music, where the core musical theme of these artists' music was about sex and getting women dance provocatively (Miller 2008; Sarig 2007). Additionally, New Orleans Bounce music took on the same theme of shaking and twerking, but with more importance on calling out wards/neighborhoods in their music. Furthermore, the growth of bounce music is due to the Black queer, trans, women rappers like Katy Redd, Big Freedia, Sissy Nobby and Magnolia Shorty, and Cheeky Blakk who disrupt the patriarchal and homophobic structures of southern hip-hop by taking on and producing bounce music (Clay 2007; Love 2017).

Atlanta hip-hop culture has a unique positionality of cultivating booty-shaking music and culture where Black woman contributed. With social gatherings like Freaknik, the Atlanta Greek Picnic and HBCU homecoming public dance performances, the congregation of young Black adults creates a southern hip-hop tradition that disrupts public spaces (Barnes 2020; Horton-Stallings 2020; Sarig 2007; Wicks 2013). Additionally, while the strip clubs in Atlanta are more private adult entertainment spaces, the attachment that hip-hop artists, labels and community members have to the strip club also caters to southern hip-hop traditions that appear in song lyrics and music videos. It is important to note that the Black women in Atlanta who occupy and perform in these spaces craft and curate a southern dance culture, and fuel the hip-hop aesthetic and economy that are not formally recognized in southern hip-hop culture.

To further extend the conversation of Black women occupying the public sphere, Black women in Atlanta have disrupted public spaces with events like Freaknik to the extent that Atlanta officials needed to rezone and control the spaces in which Black women were dancing publicly (Thompson 2007). Public youth gatherings in Atlanta take on new meaning as street-based protest where dancing and performing in the streets take on southern forms of Black expression. Many Black women displayed and performed popular

dance moves where the majority of them were of twerking and booty popping/shaking, which was described as carnivalesque in nature (Miller 2008; Thompson 2007). Many of the tensions of Freaknik or any public dancing that exposed the intimate body parts of Black women were deemed as lewd activity or self-objectification, and did not consider the cultural climate in which these Black women chose to perform (Miller 2008; Sarig 2007; Thompson 2007). The southern performances in cities like Atlanta with Freaknic or Black Bike Week in Daytona Beach, Florida cater to the southern erotic mythology that supports the lyrical and musical themes found in southern hip-hop music (Thompson 2007). The nuance in this perspective maps the geographical space that Black southerners occupy publicly, where the local response by government officials to such events is to categorize them as disorderly or criminal in action (Grem 2006; McKittrick 2006; Thompson 2007). Therefore, the public street performances of southern Black culture and hip-hop dance disregard the notion of otherness, and claim or dominate urban white neighborhoods in order to freely express modes of performance and play. In more private locations, the Atlanta strip club scene is unique, and caters to the economic impact of southern hip-hop because many southern artists like Ludacris, Future, and Young Jeezy were able to gain respect and publicity for their music from playing their early songs in the strip club. This economic tactic utilized by southern hip-hop managers like Jermaine Dupri would base the hit of the song on the reaction and participation of the stripper's willingness to dance to the artists songs (Miles 2020; Miller-Young 2014; Shaw 2020; Thompson 2007). This process was played out in the STARZ tv show *P-Valley*, where one of the main characters of the show—Lil Murda, who was an up-and-coming southern rapper—made multiple attempts to get his music played in the local Mississippi strip club in order to jump start his career. The reality is that Black women strippers in the South have a significant amount of cultural and economic capital the fuels southern hip-hop music. The music has to be "danceable" for the women to perform to, which ultimately means that they grant the opportunity for rappers to become successful. While not all southern rap artists use the strip club to pitch their music, southern rap music—and more specifically trap music—often occupies the speakers of southern strip clubs and promotes strip club-like lyrics and themes that hold cultural weight in southern hip-hop music.

Furthermore, southern Black women disrupt respectability politics through their lyrical and physical performances that occupy the public sphere. They bring wreck into southern hip-hop, where they negotiate body and sexual politics which are seen in social dances of the South like twerking, booty shaking, bounce music, trap music, bounce and Miami Bass music. It is important to note that they risk the social shame of historical stereotyping and being labeled a ratchet, a ho or a freak. However, dirty south feminism explores the capital and agency in which southern Black women and girls disrupt and redefine their sexual positionality within southern hip-hop. Black women further navigate sexuality and economic politics in the public sphere by catering to and providing the foundation for southern hip-hop music within strip clubs and street carnivals, which add to the critique and quality of southern hip-hop culture. The expressive cultures that emerge from southern hip-hop through the perspective of Black women and girls become a crucial geographical location for Black women to express their ideas and viewpoints about the conservative south they live in, and to define their own narratives, but not through the perspective of victimization, self-objectification, or shame. It is clear, once these experiences are uplifted and validated, that more will know that the South truly has something to say, and Black women rappers, performers, and dancers are saying it.

**Funding:** This research received no external funding.

**Institutional Review Board Statement:** Not applicable.

**Informed Consent Statement:** Not applicable.

**Data Availability Statement:** Not applicable.

**Conflicts of Interest:** The author declares no conflict of interest.

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
