# Peer review of "Dirty South Feminism: The Girlies Got Somethin’ to Say Too! Southern Hip-Hop Women, Fighting Respectability, Talking Mess, and Twerking Up the Dirty South"

_religions, doi:10.3390/rel12111030_

Round 1
Reviewer 1 Report
My comments are attached.

Author Response
With that said, it is in the spirit of improving the manuscript that I pose the following comments and/or questions.
- All changes from the document were changed. Thank you so much for your kind review and feedback. The track changes showed the adjustments.
Provide examples of southern songs and music videos in which Freaknik was featured.
- "This is evident in the ways Freaknik was the backdrop to a few Atlanta-based southern music videos such as and Uncle Luke’s “Work it Out” video at Freaknik 1993 and Playa Poncho and La Sno’s “Whatz Up, Whatz Up” video at Freaknik 1995. Even Atlanta producer Jermaine Dupri curated a compilation album called So So Def Bass All-Stars with various southern hip-hop artists taking on the Miami and Atlanta bass sound to produce Freaknik style music."
Are Black women allowed to change their own view of what is respectable (respectability politics) over time? What I am getting at is more life experience 7 (age) generally changes people’s views on things. So, 43-year old Karrine Steffans would generally have a very different view than 23-year old Karrine Steffans.
- This is a great question to consider. I think that there is freedom and flexibility in how women grow based on circumstances to address their sexual politic. This means that free-choice and self-reflection can look like grow to one woman vs another. I consider this an extension of Black feminist knowledge and sexual expression that supports self-awareness.
It appears like Black Hip Hop artists (Ludacris, Future, Young Jeezy and others) success is directly linked to public perception of these songs via the performance of the strippers and the response that strippers get when dancing to particular songs. Why do you believe that Black Hip Hop artists do not express public gratitude for Black women (who also happen to be female performers that catapulted their successful careers)?
- Love this question and I answered this in the revision due to another reviewer pointing this out.
- "Unfortunately, this extension of representation and idea of place lacks in the recognition of lived realities of Black women from Atlanta in hip-hop. I find this happens due to the centralization and importance of male experiences in hip-hop that is dominated by men. The lack of representation is an act of misogynoir and disregard to Black women’s marketability in hip-hop, a lack of seriousness and quality southern hip-hop music compared to east and west coast rap and the ignorance of Black women and girl’s experiences, impact and relevance to southern hip-hop culture."
Reviewer 2 Report
This is a very good, if not complicated, analysis of the narrative surround Black girls and women’s contributions to Southern hip hop. The author marries hip hop to the southern experiences of black girls and women—extending performance into among other things sex work as a collateral of the hip hop culture. The piece provides a wonderful social and cultural history as it seeks to centralize the contributions of Black girls and women. I particularly like the admonishment of their being more than mere victims of the genre.
Consider the following revisions:
Lines 130- 131 need clarity. How has “this extension “presented itself as lacking”? The author’s point is not immediately clear.
Again, lines 155-157—explain what the intersections look like.
Alternately, the piece reads like a review of the literature. This could be remedied by offering more of the author’s opinion following the presentation of other research, or at least a re-summation. In that way, it could read more like the “extended conversation” she references in line 319.
The conclusion is well done.
Author Response
This is a very good, if not complicated, analysis of the narrative surround Black girls and women’s contributions to Southern hip hop. The author marries hip hop to the southern experiences of black girls and women—extending performance into among other things sex work as a collateral of the hip hop culture. The piece provides a wonderful social and cultural history as it seeks to centralize the contributions of Black girls and women. I particularly like the admonishment of their being more than mere victims of the genre.
- Thank you so much! :)
Consider the following revisions:
Lines 130- 131 need clarity. How has “this extension “presented itself as lacking”? The author’s point is not immediately clear.
- Corrected; "Unfortunately, this extension of representation and idea of place lacks in the recognition of lived realities of Black women from Atlanta in hip-hop. I find this happens due to the centralization and importance of male experiences in hip-hop that is dominated by men. The lack of representation is an act of misogynoir and disregard to Black women’s marketability in hip-hop, a lack of seriousness and quality southern hip-hop music compared to east and west coast rap and the ignorance of Black women and girl’s experiences, impact and relevance to southern hip-hop culture"
Again, lines 155-157—explain what the intersections look like.
- Corrected; "Most importantly, Diamond and Princess made their claim to fame in southern hip-hop through the intersections of race, class and sexuality. This means considering how Diamond and Princess were one of the first to introduce people living outside the south to Black girls living, rapping, and dancing in Atlanta. With their particular urban southern style, southern rhymes, and aesthetic, Diamond and Princess represented the everyday look and experiences of Black girls from the crunk era in southern hip-hop"
Alternately, the piece reads like a review of the literature. This could be remedied by offering more of the author’s opinion following the presentation of other research, or at least a re-summation. In that way, it could read more like the “extended conversation” she references in line 319.
- Corrected and added more reflections of the literature in applicable places. I find that I struggle with finding and asserting my voice in my work.
The conclusion is well done.
- Thank you so much! :)